# Attention Schema-based Attention Control (ASAC): A Cognitive-Inspired Approach for Attention Management in Transformers

## Abstract

Attention mechanisms have become integral in AI, significantly enhancing model performance and scalability by drawing inspiration from human cognition. Concurrently, the Attention Schema Theory (AST) in cognitive science posits that individuals manage their attention by creating a model of the attention itself, effectively allocating cognitive resources. Inspired by AST, we introduce ASAC (Attention Schema-based Attention Control), which integrates the attention schema concept into artificial neural networks. Our initial experiments focused on embedding the ASAC module within transformer architectures. This module employs a Vector-Quantized Variational AutoEncoder (VQVAE) as both an attention abstractor and controller, facilitating precise attention management. By explicitly modeling attention allocation, our approach aims to enhance system efficiency. We demonstrate ASAC's effectiveness in both the vision and NLP domains, highlighting its ability to improve classification accuracy and expedite the learning process. Our experiments with vision transformers across various datasets illustrate that the attention controller not only boosts classification accuracy but also accelerates learning. Furthermore, we have demonstrated the model's robustness and generalization capabilities across noisy and out-of-distribution datasets. In addition, we have showcased improved performance in multi-task settings. Quick experiments reveal that the attention schema-based module enhances resilience to adversarial attacks, optimizes attention to improve learning efficiency, and facilitates effective transfer learning and learning from fewer examples. These promising results establish a connection between cognitive science and machine learning, shedding light on the efficient utilization of attention mechanisms in AI systems.

## 1 Introduction

Inspired by human attention, the attention mechanisms in machine learning have emerged as crucial elements in enhancing the effectiveness of AI models across a wide array of applications. Analogous to human attention, these mechanisms within AI systems, notably transformers (Vaswani et al., 2017), facilitate comprehensive global contextual understanding (each token attends to other tokens), concurrent information processing (computing weighted sums of input representations), and focus on diverse aspects (using multi-head attention), all while adapting to specific tasks through acquired attention weights. Nevertheless, human attention is a little more complicated. The Attention Schema Theory (AST) (Graziano, 2017; 2020; 2022; Wilterson & Graziano, 2021) provides a conceptual framework for elucidating how we internally regulate our focus, deliberately disregarding certain stimuli to concentrate on selected objectives.

In this work, we introduce Attention Schema-based Attention Control (ASAC), an attention abstractor and controller inspired by AST, integrated into the transformer's dot-product attention mechanism. We use Vector Quantized Variational AutoEncoders (VQVAE) as the attention controller (Van Den Oord et al., 2017). Typically used for image generation, VQVAE is adapted here to control attention with in transformers. The purpose of an attention controller is to learn how to adjust the attention for the subsequent sequential input. By merging autoencoders with Attention Schema Theory principles, it is possible to enhance the

efficiency and effectiveness of deep-learning structures. We hypothesize that the attention controller model would not only be capable of transforming data into accurate representations but also directing attention to the most relevant information, manipulating attention as needed, and making accurate predictions about future trends when plugged into the attention mechanism.

The main contributions of this paper are as follows:

1. We introduce the concept of integrating Attention Schema-based Attention Control (ASAC) into transformers using VQVAE, marking a step towards attention management in artificial networks based on consciousness theory.

2. We demonstrate improved performance across various datasets and domains, showcasing ASAC's generalization capabilities and ability to enable adaptive and dynamic attention control. Furthermore, we present encouraging outputs in multi-task scenarios, noisy and unseen dataset, thus substantiating the proposition, that AST-based attention controller facilitates adaptive and dynamic attention control.

3. We perform additional preliminary experiments demonstrating that the ASAC module not only boosts performance but also excels in various auxiliary tasks. These comprehensive adaptive capabilities include resilience to adversarial attacks, efficient learning, effective transfer of knowledge, and robust learning from fewer examples, demonstrating its broad applicability and utility.

The paper is organized as follows. We present related studies in section 2. We explain the expectations from attention schema theory concepts in deep learning models and explore the potential role of AST in transformers architecture in section 3. We elaborate our approach insection 4, followed by vision experiments and evaluation results in section 5, and NLP experiments and results in section 6. We conclude with the discussion in section 7.

## 2 Related Works

**Attention controller in transformers:** Several deep learning models have attempted to draw inspiration from theories of consciuosness in humans, with a focus on representation learning. The Global Workspace theory (Newman & Baars, 1993) proposes a system that distributes information among specialized modules to enhance cognition and awareness.

Deep learning models inspired by the concept of a global shared workspace include architectures that utilize a common representation for multiple input modalities (Devillers et al., 2024). These models leverage specialized systems for each modality, with latent representations encoded and decoded within a shared workspace (Goyal et al., 2021; VanRullen & Kanai, 2021). (Piao et al., 2020) explores the Hybrid CNN and Adaptive DenseNet models inspired by a global shared workspace in the Flexible Parameter Sharing Networks approach. (Peis et al., 2023) presents a novel deep generative model based on non-i.i.d. variational autoencoders that capture global dependencies among observations in a fully unsupervised fashion and study the ability of the global space. The shared workspace facilitates communication among different specialized modules, akin to a bandwidth-limited communication channel in cognitive science (Franklin et al., 2007; 2012). Furthermore, the proposal of a global latent workspace (GLW) (VanRullen & Kanai, 2021) through unsupervised neural translation between multiple latent spaces aligns with the Global Workspace theory for creating higher-level cognition. Additionally, the LIDA cognitive architecture (Franklin et al., 2007; 2012) implements the Global Workspace Theory conceptually and computationally, emphasizing the importance of modeling high-level cognitive processes inspired by biological cognition. These models, inspired by the Global Workspace Theory, represent significant steps toward creating biologically inspired cognitive architectures that can mimic the human mind's ability to process and integrate information across various domains and modalities.

**Attention Schema Theory:** In cognitive neuroscience, the Attention Schema Theory (AST) is a significant framework that proposes the brain develops a schema to manage attention efficiently (Graziano, 2017; 2020; 2022; Wilterson & Graziano, 2021). This schema predicts and characterizes attention focus independently

from directing attention. AST highlights the benefits of having an attention schema, especially in enhancing visuo-spatial attention control (Liu et al., 2023). It also links subjective awareness to attention modulation, emphasizing awareness's role in attention governance. Wilterson & Graziano (2021) support the attention schema's role in improving social intelligence and agent coordination.

Attention schema is viewed as an internal model enabling the brain to abstractly represent, manipulate and forecast attention dynamics (Liu et al., 2023; Graziano, 2017; Graziano & Webb, 2015; van den Boogaard et al., 2017). Similar to the body schema for movement control, this model aids top-down attentional control and task performance. AST suggests that lack of awareness may compromise attention control, showing the interplay between attention and consciousness. Simulations based on AST illustrate how attention control can impact cognitive functions.

Building on these insights, our paper introduces an architecture inspired by the Attention Schema Theory (AST), integrated within transformer models. We demonstrate how this novel integration enhances the performance and adaptability of transformers, both in vision tasks and preliminary experiments with NLP tasks. Our findings suggest that applying cognitive theories like AST to AI models can significantly improve their efficiency and robustness, opening new avenues for developing more sophisticated and cognitive-inspired AI systems.

## 3  Attention Schema-based Attention Control in Deep Neural Networks

**Attention Schema and Integration of Human-Like Attention Mechanisms in AI:** AST proposes that the brain creates an "attention schema" reflecting our focus and related cognitive activities. This schema supports complex cognitive tasks like reasoning and decision-making. AST explains that just as the brain creates a simplified model of the body to control movements, it also makes a model of attention to manage focus and cognitive processes. While humans naturally manage attention, replicating this in AI systems is challenging.

Most sophisticated AI systems employ conventional attention mechanisms (Liu et al., 2019; Yang et al., 2019; Dai et al., 2019; Sanh et al., 2019; Lewis et al., 2019) or algorithmic adjustments to enhance performance (Child et al., 2019; Kitaev et al., 2020; Beltagy et al., 2020; Wang et al., 2020), yet replicating the nuanced attention control observed in humans remains a work in progress. In this paper, we present a cognitive-inspired approach, ASAC, to efficiently control the attention in AI models.

**Examining the Potential Roles of Attention Schema in Transformers:** Attention Schema Theory (AST) posits that the brain creates a simplified model of its attention system to manage cognitive resources efficiently (Graziano & Webb, 2015). This internal schema helps predict and adjust attention for various tasks. For instance, when repeatedly performing similar actions like picking up objects, the brain reuses and refines this map to distribute resources efficiently. Conversely, when faced with new challenges, such as solving a puzzle, the brain uses the attention schema to adapt the attention to meet changing demands. We explore whether the ASAC model develops a similar attention schema to allocate cognitive resources effectively. In standard transformers (Vaswani et al., 2017), the attention blocks learn weights during training that determine how inputs are processed. We hypothesize that multiple distinct attention patterns emerge, effectively clustering inputs into subsets, each processed according to its asSOCiated attention configuration. To capture this behavior, we introduce a higher-level structure that maps attention scores to a finite set of latent representations. Specifically, we apply a Vector Quantized Variational Autoencoder (VQ-VAE) to the attention scores, compressing them into discrete codes. These codes act as attention schemas, consistent with AST, and enable more robust associations between inputs and their corresponding attention patterns. This compression improves noise tolerance and stability when processing out-of-distribution data, as demonstrated in our experiments.

AST suggests the brain strategically distributes cognitive resources and adapts to changing environments. Similarly, we propose that ASAC can dynamically allocate attention, enhancing resilience to noise and unfamiliar data. Instead of fixed attention patterns, these models should flexibly adapt to diverse inputs. We demonstrate this through experiments on noisy and out-of-distribution datasets. The human brain transitions seamlessly between tasks due to cognitive flexibility. We propose that AST-based models can

similarly allocate attention across tasks, dynamically adjusting based on requirements rather than treating jobs in isolation. Additionally, AST-based model should optimize the resource allocation for solving similar or dissimilar tasks, just as human brain does. Similar tasks should re-use the cognitive resources while dissimilar tasks may distribute the resources to adapt to the needs. We demonstrate this capability in multi-task scenarios.

Furthermore, preliminary studies indicate that the ASAC module enhances robustness and adaptability in various contexts. ASAC improves resilience to adversarial attacks by dynamically adjusting attention patterns to mitigate the impact of misleading inputs. Additionally, similar to how the brain efficiently learns new skills with minimal repetition, ASAC enhances learning efficiency, enabling models to achieve better performance with fewer training epochs. In transfer learning scenarios, akin to how humans apply knowledge from one domain to another, ASAC facilitates the effective transfer of knowledge, promoting quicker adaptation to new tasks. Moreover, reflecting the brain's ability to learn effectively from limited information, ASAC excels in scenarios with limited data, focusing on the most relevant information to learn from fewer examples. These capabilities underscore the comprehensive adaptive nature of ASAC, making it an invaluable tool for advancing AI model performance across a wide range of challenges.

## 4 Proposed Method

In this section, we present the proposed method and ways we have integrated the ASAC module in different scenarios to test its performance.

### 4.1 Attention Controller

The AST would inspire a framework that facilitates abstraction, manipulation, and prediction. Abstraction involves creating a simplified attention schema to guide attention on vital information, improving model performance. Manipulation entails fine-tuning attention to emphasize important details, and ensuring relevant data is prioritized. Prediction involves the model's forecasts about future attention allocation based on processed data, allowing it to foresee trends and enhance predictive accuracy. This triad of capabilities is central to our model's design, enabling it to efficiently process and anticipate information.

We incorporate an attention controller into the transformers architecture (Dosovitskiy et al., 2020), using Vector Quantized Variational Auto-encoder (VQVAE) (Van Den Oord et al., 2017) to embody the concepts of abstraction, manipulation, and prediction. VQVAE reflects Attention Schema Theory by abstracting data into a discrete space, thereby creating an attention schema. It manipulates attention through vector quantization, learning from data to focus on specific elements, and its decoder predicts by reconstructing inputs from these discrete latent representations.

We show the overall architecture in Figure 1. We plug the attention controller (VQVAE) into the attention mechanism of the transformer, where the attention scores are calculated. The input involves the scaled dot product of the query and keys, which is then fed into the encoder. Both the encoder and the decoder consist of two linear layers separated by a LeakyReLU activation function. A discrete embedding vector matrix, referred to as a codebook, is randomly initialized at the start. The distance between the codebook vectors and the encoding output is calculated, and the closest vectors to the encoder input are indexed from the codebook, and exponential moving average for these vectors is calculated to update the codebook while learning. The chosen vectors from the codebook serve as the input to the decoder. Throughout the training phase, adjustments are made to the codebook to reduce the reconstruction error between the input and output. The output is the reconstructed scaled dot product, via which we compute the final attention. We also add residual connections between the input and the output attentions scores to facilitate stable training. For this, we simply add the reconstructed scaled dot product to the original input scaled dot product.

We employ a composite loss function for the model, comprising three distinct types of losses. The reconstruction loss $L_{\text{recon}}$, is the mean squared error loss between the reconstructed attention and the original

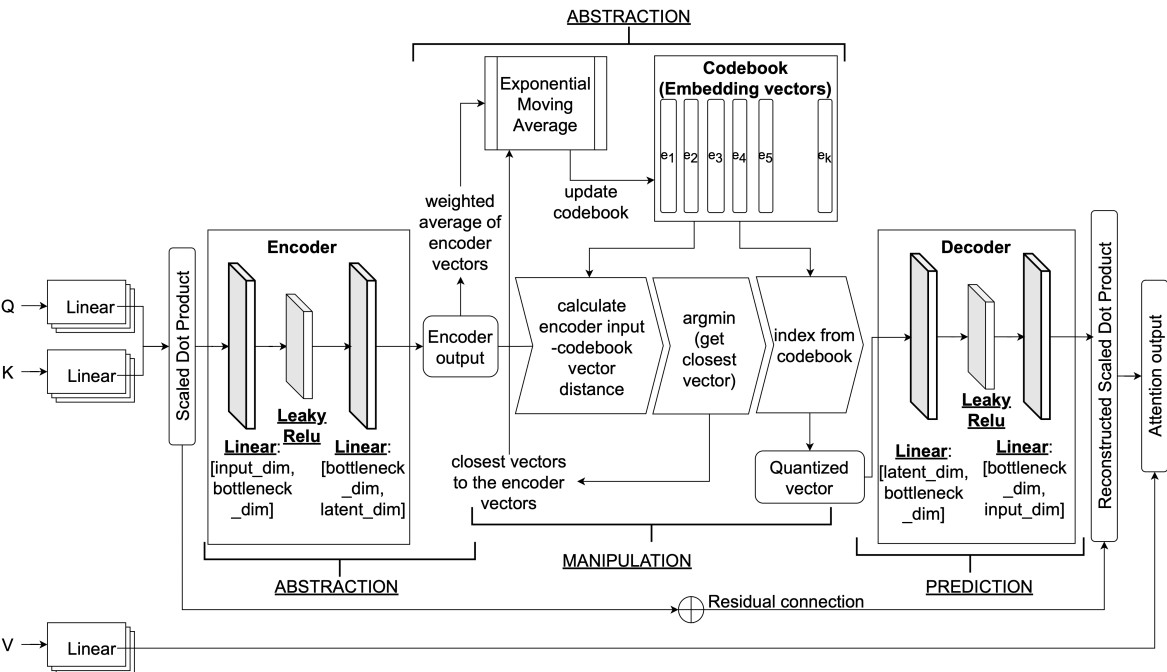

Figure 1: Proposed architecture of ASAC. The VQVAE module is inseRTEd in the attention mechanism and reconstructs the scaled dot product between queries and keys.

attention. The reconstruction loss is defined by the following equation:

$$L_{\text{recon}} = \frac{1}{N} \sum_{i=1}^{N} \left( A_{\text{original}}^{(i)} - A_{\text{reconstructed}}^{(i)} \right)^2 \tag{1}$$

where, $A_{\text{original}}^{(i)}$ and $A_{\text{reconstructed}}^{(i)}$ are the original and reconstructed attention matrix, $N$ is the total number of scores calculated in one forward-pass. The VQVAE attention controller has a vector quantization loss (Dosovitskiy et al., 2020) specific to the VQVAE architecture. It ensures that the latent representations (the encodings) match the closest vector in a learnable discrete codebook. The VQ loss can be broken down into two components: the codebook loss and the commitment loss, which help maintain a balance between the flexibility of the codebook and the stability of the encodings. The VQ loss is defined as:

$$L_{VQ} = \|\text{sg}[z_e(x)] - e\|_2^2 + \beta\|z_e(x) - \text{sg}[e]\|_2^2 \tag{2}$$

where, the first term is the codebook loss and the second term is the commitment loss (Dosovitskiy et al., 2020). The codebook loss ensures that the chosen codebook vectors (the embeddings ($e$)) are close to the encoder outputs ($z_e(x)$)). ($sg$) represents the stop-gradient operator, meaning no gradient flows through whatever it's applied on. This loss moves the embedding vectors ($e$) towards the encoder output ($z_e(x)$)). The commitment loss encourages the encoder outputs to commit to a codebook vector, preventing them from drifting too far during training, ($\beta$) is a hyper-parameter that balances the importance of the commitment loss relative to the codebook loss. For experiments in this paper, $\beta$ is set to 1.

The task loss, $L_{\text{task}}$, as the name suggests is the overall loss function specific to the task being solved by the model. It can be binary cross-entropy loss or multi-class cross-entropy based on the task. The cross-entropy loss for multi-class classification is given by:

$$L_{\text{task}} = -\sum_{c=1}^{M} y_{i,c} \log(p_{i,c}) \tag{3}$$

where, $(M)$ is the number of classes, $(y_{i,c})$ is a binary indicator of whether class $(c)$ is the correct classification for observation $(i)$, and $(p_{i,c})$ is the predicted probability that observation $(i)$ is of class $(c)$. Similarly, if the task is binary classification, then the loss is given as:

$$L_{\text{task}} = -\left[y_i \log(p_i) + (1 - y_i) \log(1 - p_i)\right] \tag{4}$$

where, for the $(i^{th})$ sample, $(y_i)$ is the true label (0 or 1), and $(p_i)$ is the predicted probability of the observation being in class 1.

The overall loss is calculated as:

$$L_{\text{final}} = L_{\text{task}} + \lambda \left(L_{\text{recon}} + L_{VQ}\right) \tag{5}$$

## 4.2 Attention Control on Multi-task Scenario

The Attention Schema Theory (AST) suggests that cognitive flexibility is essential for smooth transitions between tasks within the human brain. Similarly, a model based on AST should be capable of dynamically transitioning between tasks by adjusting the allocation of attention according to the requirements of each task. A core part of our study involves evaluating our model in scenarios involving binary or multiple tasks, which can range from simple binary or multi-class classification to more complex multi-class multi-label classification.

To perform this experiment, we incorporate unique task identifiers (Task IDs) within the dataset to differentiate between various tasks. Although the dataset remains constant, the corresponding labels may vary depending on the task identifiers. During the classification trials involving an attention controller, the original input images are transformed into a sequence of distinct patches. These patches are then conveRTEd into patch embeddings, with positional embeddings computed to form the final input sequence fed into the model. In a multi-task setting, we augment this input sequence with a task embedding before processing it through the model. We tried the following variations for task information processing:

- Including the Task ID information in the input sequence: This is like priming the attention schema with the task to be solved from the beginning.

- Including the Task ID information to the VQVAE decoder: This concept is aligned with the notion that the attention schema is adaptable based on the task requirements during the interpretative phase.

- Including the Task ID information to both the input sequence and the decoder: This represents holistic integration where attention schema is continuously informed and adjusted based on the task information from initial processing to final decision-making.

## 4.3 Attention Control on Noisy and Out-of-Distribution Datasets

The attention schema enables prioritizing and focusing on the most relevant information. We hypothesize that the AST-based model should also be able to generate this selective attention on data features to attenuate irrelevant or misleading signals. Consequently, the attention schema should enhance the generalization power of the model. To validate this proposition, we assess our model using datasets containing noise and data points outside the distribution while being trained on the original, non-noisy dataset.

## 4.4 Assessing Versatility and Adaptability of the Attention Controller

The attention schema flexibly adapts to diverse inputs, and due to the discretization of information using a codebook, it enhances resilience to subtle changes in the inputs, such as adversarial attacks. ASAC should also be adept at learning faster with fewer samples by optimizing attention and focusing on relevant information. Additionally, it should facilitate the transfer of knowledge from one task to another by adapting attention patterns. We conduct preliminary investigation on versatility and adaptability of ASAC.

### 4.5 Attention Control for Optimizing Resource Distribution

AST posits that the brain creates a simplified model of its attention system to manage cognitive resources efficiently (Graziano & Webb, 2015). This internal schema helps predict and adjust attention for various tasks. For instance, when repeatedly performing similar actions like picking up objects, the brain reuses and refines this map to distribute resources efficiently. Conversely, when faced with new challenges, such as multi-tasking, the brain adapts the schema to meet changing demands. We explore whether the ASAC model develops a similar attention schema to allocate cognitive resources effectively.

## 5 Experiments on Vision Datasets

In this section, we describe the experiments we conducted and the datasets used. Each subsection contains the results for those experiments.

### 5.1 Attention Controller Performance on Classification

To evaluate the effectiveness of ASAC on transformers, we conducted classification experiments on various vision datasets. For the vision experiments, we used vision transformers[1] without the ASAC module as a baseline. We compared this baseline to a model with the ASAC module integrated into all layers. Both models were trained from scratch on the datasets, with hyper-parameters and training conditions detailed in the Appendix.

#### 5.1.1 Datasets

The vision classification tasks include: 1) binary classification on Triangles and Polygons, 2) multi-class classification on FashionMNIST, SVHN, TypefaceMNIST, CIFAR10, CIFAR100, Tiny Imagenet, and Places365, 3) multi-label classification on CelebA, and 4) VQA on the Sort-of-Clevr dataset. Out of these, CIFAR10, CIFAR100[2], FashionMNIST[3], TypefaceMNIST[4], SVHN[5], Tiny Imagenet (Deng et al., 2009), Places365-Standard (Zhou et al., 2017a;b) and CelebA (Liu et al., 2015) are standard classification datasets.

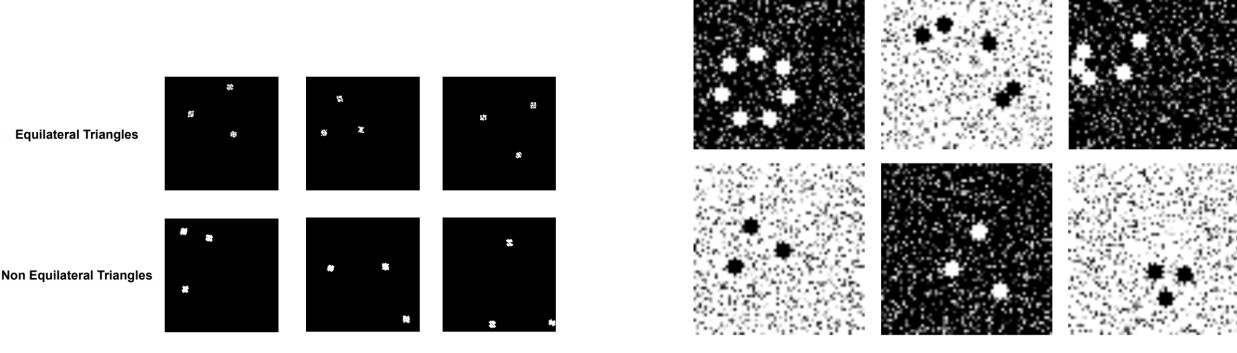

(a) Samples from Triangle dataset.      (b) Samples from Polygon dataset.

Figure 2: Samples from different datasets.

We explain the rest of the datasets below:

---

[1]https://huggingface.co/docs/transformers/main/en/model_doc/vit
[2]https://www.cs.toronto.edu/~kriz/cifar.html
[3]https://docs.pytorch.org/vision/stable/generated/torchvision.datasets.FashionMNIST.html
[4]https://www.kaggle.com/datasets/nimishmagre/tMNIST-typeface-MNIST
[5]http://ufldl.stanford.edu/housenumbers/

- Triangles: We adopt this dataset from (Goyal et al., 2021; Ahmad & Omohundro, 2022), containing images of three white dot clusters on a black background. The task is to predict if the clusters form an equilateral Triangle.

- Polygons: This variation of the Triangles dataset includes images with three or more clusters. The task is to identify regular Polygons with equal side lengths, similar to equilateral Triangles but with added complexities like background noise and alternating hues. We train the model on images with 3, 4, or 8 vertices and 5% noise, and test on images with 5, 6, or 7 vertices and 25% noise. Images may also be negative, with switched foreground and background colors. We show the samples from Triangles and Polygons data in Figure 2.

- Sort-of-Clevr: Sort-of-Clevr (SOC)[6] (Goyal et al., 2021; Santoro et al., 2017) is a dataset for testing reasoning abilities, similar to CLEVR (Johnson et al., 2017). It includes images of 2D objects and questions about their properties and relationships. The dataset has two tasks: non-relational reasoning [SOC (norel)], focusing on a single object's properties like shape and location, and relational reasoning [SOC (rel)], involving relationships between objects like proximity or counting similar shapes.

More information on datasets is available in Appendix.

### 5.1.2 Results

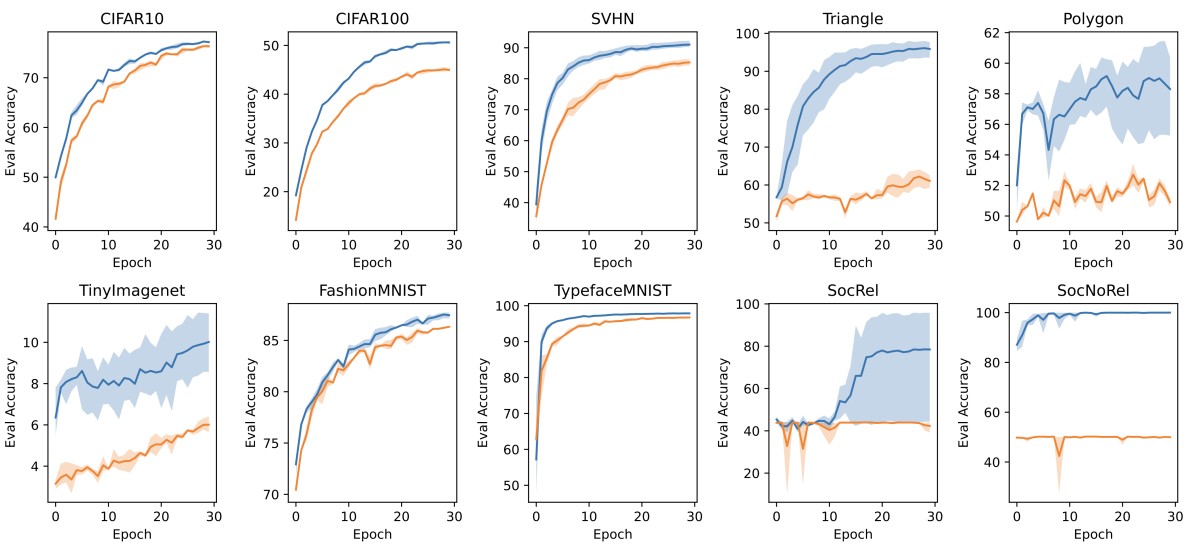

Figure 3: Classification results on vision datasets. Blue and orange represents ASAC and baseline, respectively.

We present the performance of the ASAC architecture in vision classification experiments, as illustrated in Figure 3 and Figure 4. For all vision experiments in this paper, we trained the model from scratch, without any pre-training, to assess the effectiveness of the attention controller in feature extraction and task understanding across three different seeds, and we aggregated the results. The classification results for all datasets, except CelebA and Places365, are shown in Figure 3. The results indicate that ASAC outperforms baseline transformers in all experiments and facilitates faster learning compared to the baseline transformers.

Figure 4 (first three plots) displays the results for the multi-label classification experiment on the CelebA dataset, demonstrating that ASAC achieves superior performance compared to the baseline. The last plot of Figure 4 shows the results for the Places365 dataset. Here, ASAC exhibits faster learning and reaches higher accuracy quicker than the baseline. However, after several epochs, we observe over-fitting in both ASAC

---

[6]https://github.com/RishikMani/Sort-of-Clevr

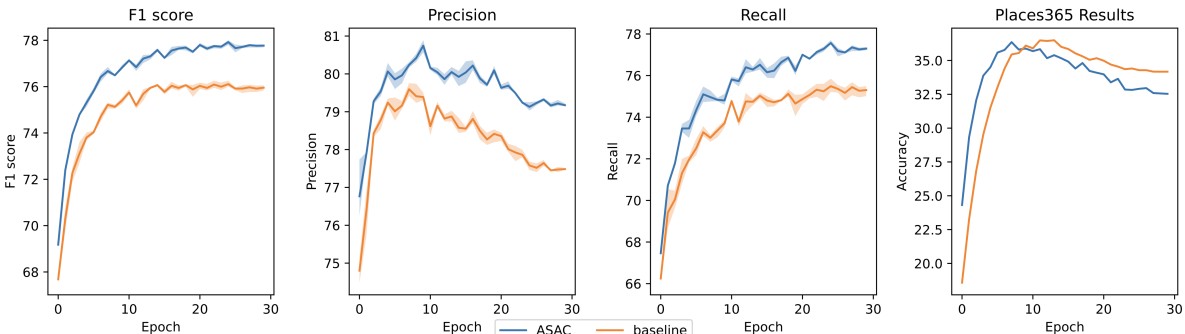

Figure 4: Classification results on CelebA and Places365.

and the baseline models. Although we show results for 30 epochs for consistency, early stopping would have halted training within 10 epochs for Places365 to prevent over-fitting. Places365 is a large dataset requiring significant computational resources, and due to cost constraints, we could not run multiple seeds or test various hyper-parameter settings.

## 5.2 Multi-task Handling

In these experiments, we assess the efficacy of the attention controller in comprehending various tasks and generating predictions according to the task at hand.

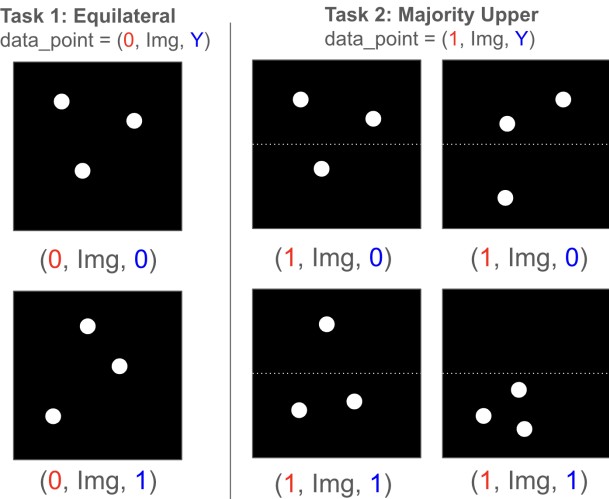

Figure 5: Multi-task scenario with Triangles data. Left side shows task 1: detecting equilateral Triangles and right side shows task 2: detecting majority upper points. First entry in the data points are the task ID.

### 5.2.1 Datasets

We asses the efficacy of the attention controller in comprehending various tasks using the following datasets:

- Sort-of-Clevr: We use the same dataset as before, posing relational and non-relational tasks separately.

- Triangles: Using the same images as before, we create labels for two tasks: identifying equilateral Triangles and determining if most cluster points are in the upper half. We show a sample in Figure 5.

Table 1: Results of the multi-task experiments. Due to cost constraints for computational resources, we run Places365 with only one variant where the taskID is added to the VQVAE input and decoder. Bold values represent the best performance.

| Metrics ↓ | Model → | Baseline | ASAC | | |
| | TaskID in → | input | input | decoder | input&decoder |
| | Dataset ↓ | | | | |
| Accuracy | SOC | $63.15 \pm 7.39$ | $67.08 \pm 2.84$ | $\mathbf{73.23 \pm 6.4}$ | $71.09 \pm 4.35$ |
| | MNIST | $96.78 \pm 0.4$ | $97.17 \pm 0.13$ | $97.04 \pm 0.26$ | $\mathbf{97.35 \pm 0.32}$ |
| | CIFAR10-SVHN | $85.94 \pm 1.72$ | $\mathbf{86.58 \pm 0.64}$ | $84.25 \pm 0.28$ | $86.56 \pm 0.63$ |
| | Triangles | $78.66 \pm 0.83$ | $\mathbf{96.71 \pm 1.28}$ | $96.01 \pm 1.39$ | $94.39 \pm 3.2$ |
| | ODIR-5K | $83.44 \pm 0.01$ | $\mathbf{83.46 \pm 0.01}$ | $75.60 \pm 5.28$ | $79.32 \pm 5.24$ |
| Precision | Places365 | 60.46 | - | - | **61.03** |
| Recall | | 54.04 | - | - | **54.56** |
| F1 | | 56.25 | - | - | **56.82** |

- MNIST: MNIST[7] is classic ML dataset for which we create labels for two tasks: distinguishing odd from even numbers and identifying prime versus composite numbers.

- Ocular Disease Intelligent Recognition: ODIR-5K[8] (Prabhakaran et al., 2024) is real-world ophthalmic disease dataset with 8 labels. We pose it as a multi-task problem, with each task being a binary classification of whether an image represents a specific disease.

- Places365: Using the same images, we create three tasks: single-label classification, and multi-label classification into level-1 and level-2 hierarchy labels (Zhou et al., 2017a;b).

- CIFAR10-SVHN: We mix the CIFAR10 and SVHN dataset and the task is to label the images of CIFAR10 or SVHN based on the task ID input.

More information on the datasets is available in Appendix.

### 5.2.2 Results

Table 1 illustrates that the AST-based attention controller exhibits superior capability in adjusting the attention schema within a multi-task environment when compared to the model lacking the controller. In these experiments, we run three different seeds and present aggregate results. We see that the best-performing model among different datasets is a different variant of the model. In datasets like ODIR-5K, Triangles and CIFAR10-SVHN, the taskID in input performs the best, which means task-specific information (TaskID) provided at the input level helps the model to quickly adapt and focus on the relevant features for each task. We observe that task information in the decoder helps in Sort-of-Clevr. By sending the TaskID to the decoder, the model can leverage the task-specific information at a later stage, allowing the encoder to focus on extracting general features without being biased by the task. For MNIST, we see that the best performing model is when the task information is present in the input as well as the decoder, meaning that the model is able to make better decisions when the model is continuously informed and adjusted based on the task information. Since Places365 is a huge dataset and the tasks vary significantly (single-label and multi-label classification in different tasks), we assumed that task information in both the input and the decoder will help solve this task as the model can retrieve features based on the task and also conditioned to predict the final output based on the task information. We see ASAC performing better than the baseline in this setting for Places365.

---

[7]https://huggingface.co/datasets/ylecun/MNIST
[8]https://www.kaggle.com/datasets/andrewmvd/ocular-disease-recognition-odir5k

### 5.3 Generalization Power

In these experiments, we train the model with non-noisy original datasets and test on noisy and out-of-distribution (OOD) datasets to check the generalization power of the attention controller.

#### 5.3.1 Datasets

We test the generalization power of our model two noisy datasets: CIFAR-10C[9] and Tiny Imagenet-C[10]. Additionally, we test ASAC on the following OOD datasets:

- Triangles-OOD: We create test images with different sizes of clusters. We also vary the shape of clusters to circles, squares, or Triangles and the shape can be empty or filled. We train using the original (filled, circle, constant size clusters) Triangles dataset.

- Polygons-OOD: The train set contains 3,4,8 vertices with colored noise on 5% of the background. The test set contains 5, 6, and 7 vertices with colored noise on 25% of the background area.

#### 5.3.2 Results

The results for generalization experiments are shown in Figure 6. There is a marginal increase in performance with attention controller on corrupted Tiny Imagenet dataset. Since this is a challenging dataset, the overall accuracy obtained by both the baseline and VQVAE models is relatively low. The ASAC model achieves higher accuracies in corrupted CIFAR datasets. Furthermore, we also see significant improvement in the OOD experiments with Triangles and Polygons data.

### 5.4 Versatility and Adaptability

We conduct preliminary investigations to evaluate the versatility and adaptability of the model with the attention controller. Our experiments focus on four tasks: adversarial attacks, transfer learning, few-shot learning, and learning efficiency.

For the adversarial attacks experiments, we train the model for 20 epochs on the original dataset and test it on perturbed images. These attacked images, generated using the Fast Gradient Sign Method (FGSM) (Goodfellow et al., 2014) and Projected Gradient Descent Method (PGDM) (Ayas et al., 2022), are slightly altered to cause the model to misclassify them. FGSM calculates the gradient of the model's loss with respect to the image and adds a scaled version of this gradient to the original image, where the scale factor, $\epsilon$, controls the perturbation magnitude. PGDM iteratively applies small perturbations, recalculating the gradient at each step to maximize the model's error.

For transfer learning experiment, we pre-train the model on one dataset for 10 epochs and fine-tune on another (related) dataset for 5 epochs. For the few-shot experiments, we train the model for 10 epochs on one dataset and then use different percentages of another data to do few-shot training. For testing the learning efficiency, we train the model using different percentages of data for 25 epochs and test on the original test set of the same dataset. We run the experiments for three different seeds and show the aggregated results. For these experiments, we used CIFAR10, CIFAR100, FashionMNIST, SVHN dataset, described in subsubsection 5.1.1.

#### 5.4.1 Results

We show the results for FGSM and PGDM adversarial attacks in Figure 7 and Figure 8. In the FGSM adversarial attacks, the performance varies. Specifically, for CIFAR10, ASAC consistently underperforms compared to the baseline across all $\epsilon$ values. Conversely, ASAC demonstrates superior performance for FashionMNIST across all $\epsilon$ values. For CIFAR100 and SVHN, ASAC exhibits better performance at smaller $\epsilon$ values; however, as $\epsilon$ values increase, indicating stronger attacks, the baseline begins to outperform ASAC.

---

[9]https://github.com/hendrycks/robustness
[10]https://github.com/hendrycks/robustness

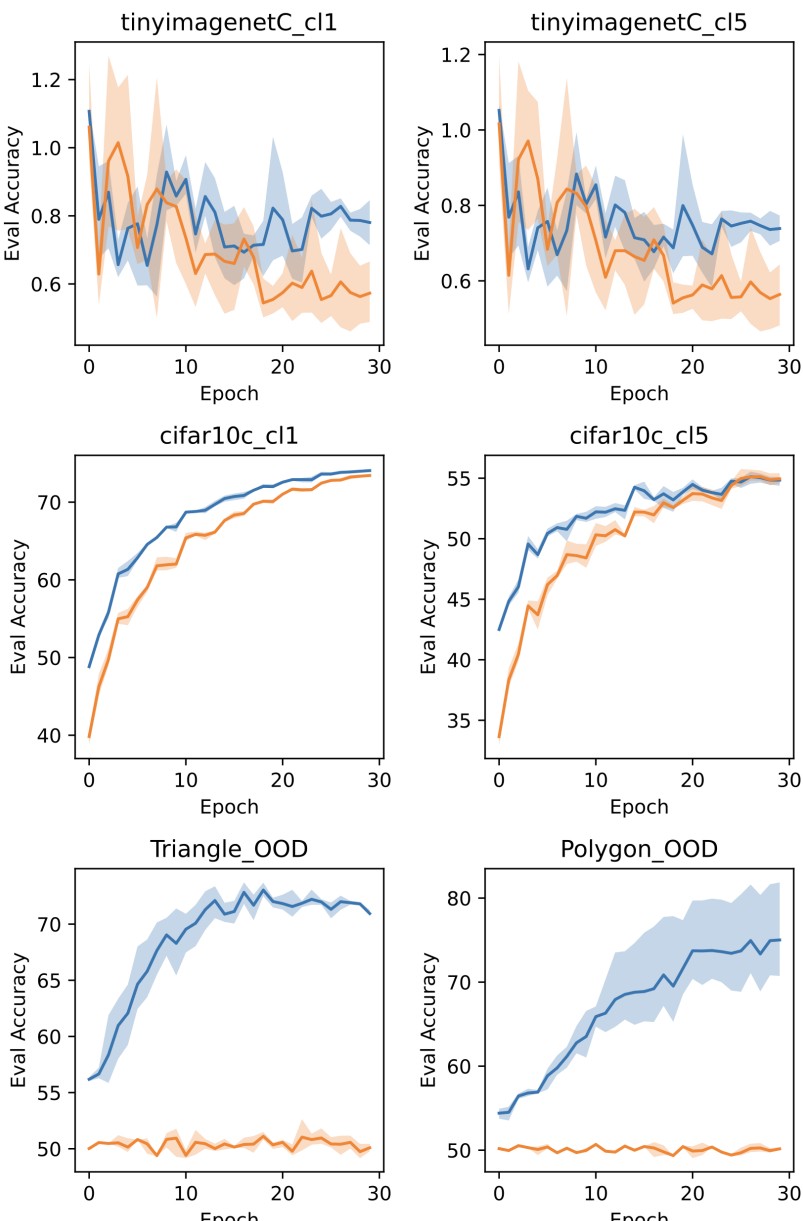

Figure 6: Generalization results on noisy and OOD datasets. First and second row are the results of corrupted TinyImagenet and CIFAR10, respectively. Last row shows the results for OOD experiments for Triangles and Polygons. Blue and orange color represnts ASAC and baseline, respectively.

In contrast, the results for PGDM attacks show that ASAC consistently surpasses the baseline across all datasets and $\epsilon$ values.

The mixed results for FGSM attacks can be attributed to the nature of the datasets and the varying complexity of adversarial examples. For CIFAR10, the simpler architecture of the baseline might be more resilient to minor perturbations, leading to better performance at all $\epsilon$ levels. For FashionMNIST, the inherent robustness of ASAC's attention mechanisms likely provides it with an advantage over the baseline, enabling it to better handle adversarial examples. In the cases of CIFAR100 and SVHN, ASAC's superior performance at lower $\epsilon$ values could be due to its ability to capture complex features effectively. However, as the attack strength increases, the baseline's more straightforward architecture might become advantageous

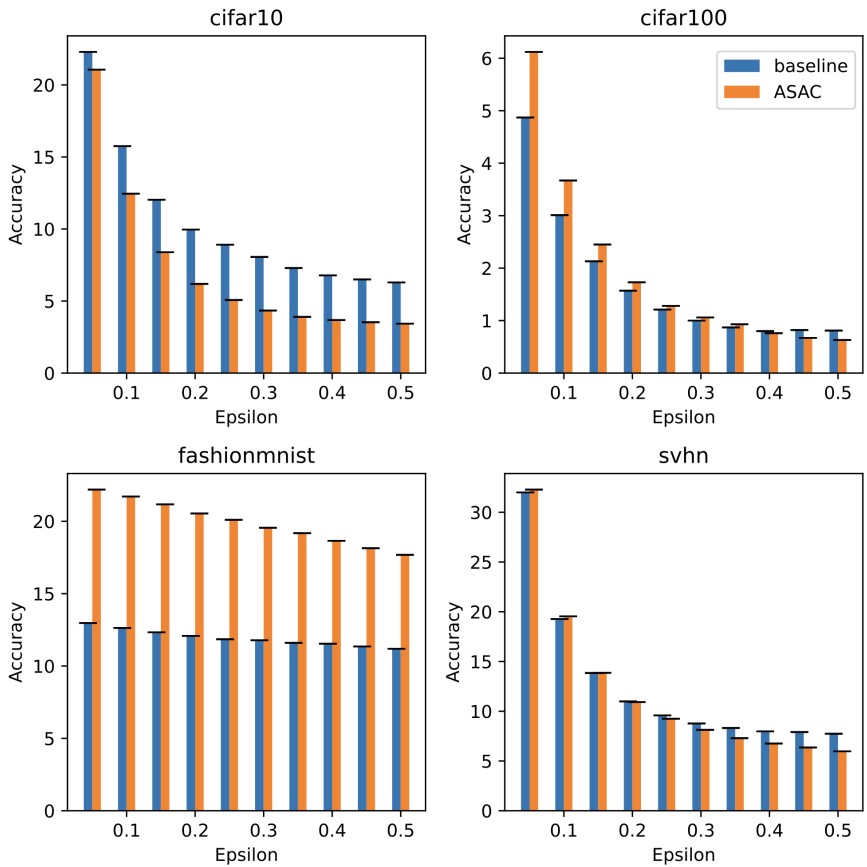

Figure 7: Adversarial attacks results for FGSM attacks on different datasets.

by avoiding overfitting to perturbed data, thus outperforming ASAC in higher $\epsilon$ scenarios. Regarding PGDM attacks, ASAC's consistent outperformance across all datasets and $\epsilon$ values highlights its robust design. The combination of improved feature extraction and attention mechanisms likely equips ASAC with enhanced capabilities to mitigate the effects of stronger, iterative adversarial attacks compared to the baseline.

The results of our transfer learning experiments are illustrated in Figure 9. We observe that ASAC consistently outperforms the baseline when trained on one dataset and fine-tuned on another. This superior performance can be attributed to ASAC's ability to effectively utilize attention patterns during the transfer of knowledge. Its design allows it to selectively focus on crucial aspects of the pre-training data, which enhances its generalization capability. Consequently, ASAC not only learns robust representations from the initial dataset but also successfully adapts and fine-tunes these representations to excel on the target dataset.

We show the results of few-shot learning in Figure 10. We observe that ASAC demonstrates enhanced accuracies when pre-training is performed on CIFAR100 and fine-tuning on CIFAR10, as well as in another experiment where pre-training is conducted on SVHN followed by few-shot fine-tuning on CIFAR10. These improvements can be attributed to the complementary nature of the datasets and the effective transfer of learned features, allowing ASAC to leverage its attention mechanisms efficiently.

For the former combination, the shared object-centric nature of CIFAR100 and CIFAR10 allows ASAC to apply its learned representations effectively, resulting in improved performance.

In the case of pre-training on SVHN and fine-tuning on CIFAR10, the larger number of training samples in SVHN may contribute to the development of more robust initial features. However, this scenario may not be as straightforward, and the effective transfer could also be attributed to the general adaptability of ASAC's

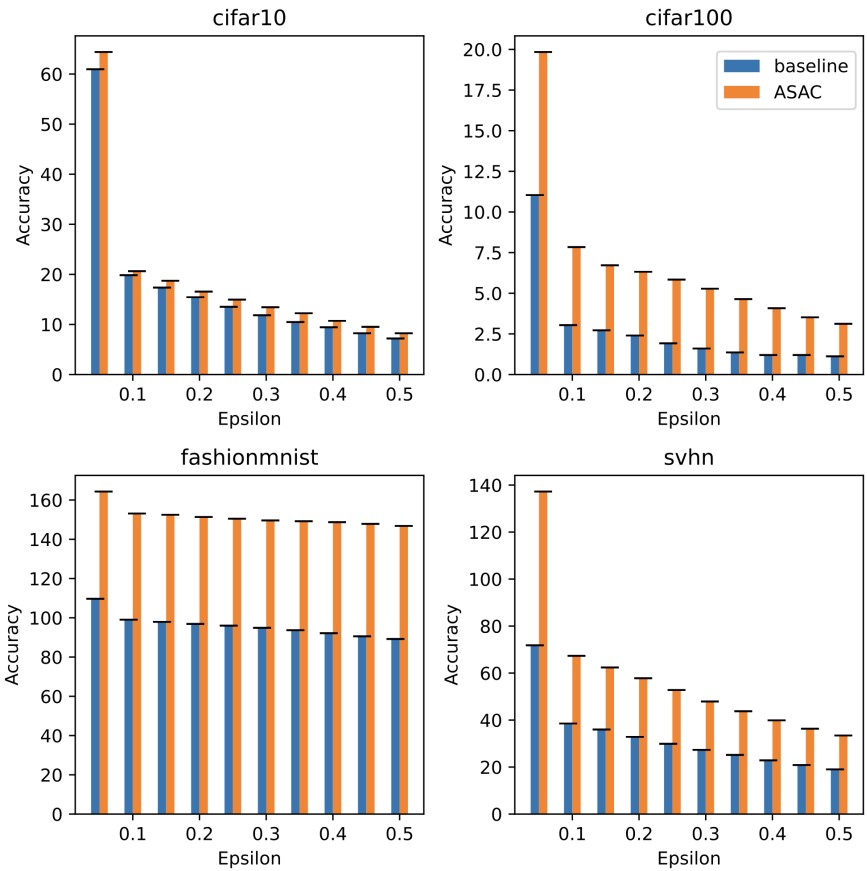

Figure 8: Adversarial attacks results for PGDM attacks on different datasets.

attention mechanisms, which manage to refine and adapt learned features efficiently to the new context presented by CIFAR10.

Conversely, for the scenarios where pre-training is done on CIFAR100 or CIFAR10 and few-shot fine-tuning on SVHN, we observe mixed results across different few-shot percentages of the data. This variability can be explained by the differences in data distribution and feature complexity between these datasets. While CIFAR datasets focus more on varied objects, SVHN is centered around digit recognition in different contexts, leading to potential mismatches in learned features during transfer learning. Consequently, the effectiveness of knowledge transfer fluctuates, resulting in inconsistent performance for different percentages of the few-shot data.

Our findings on learning efficiency experiments, depicted in Figure 11, reveal that ASAC consistently outperforms the baseline across all datasets and varying data percentages used for training the models. ASAC excels in rapid learning with fewer samples by optimizing attention and effectively focusing on relevant information. This efficiency allows ASAC to achieve higher accuracies compared to the baseline, starting from the very initial epochs.

## 5.5 Allocation of Cognitive Resources

In the VQVAE architecture, the codebook is crucial as it represents part of the attention schema, enabling efficient allocation of computational resources. We test this by analyzing the ODIR-5K and CIFAR10-SVHN experiments in multi-task scenarios, whose results are presented in subsubsection 5.2.2. Specifically, we calculate and analyze the usage of distinct codes from the codebook for different tasks to demonstrate effective resource allocation.

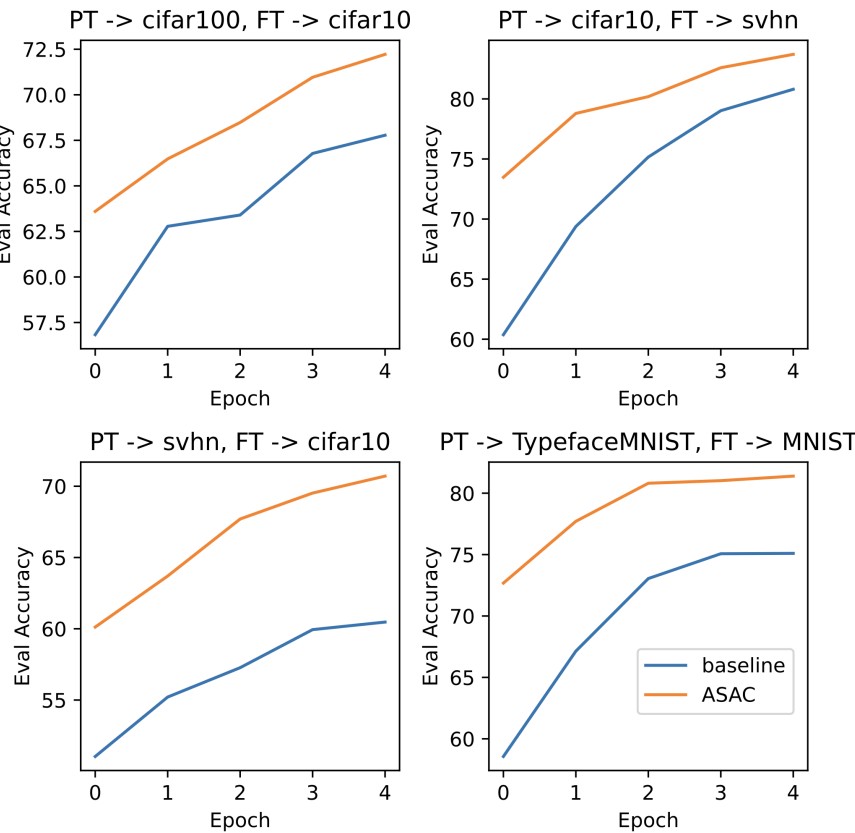

Figure 9: Results of transfer learning experiments. PT and FT stands for datasets used for pre-training and fine-tuning.

### 5.5.1 Results

We assess codebook usage by counting the frequency of specific codes for each task and calculating the Kolmogorov-Smirnov (KS) test *p-value*. For the CIFAR10-SVHN dataset, ASAC produces a *p-value* of 0.0052, indicating significant differences in code usage between tasks. This suggests the model uses distinct codes to label images in CIFAR10 (everyday objects) and SVHN (house numbers), given their differing image types. For ODIR-5K, we calculate pairwise KS-Test *p-values* for 8 tasks shown in Figure 12. All pairwise *p-values* fall between 0.21 and 1.0, with most above 0.7. Since no value for ODIR-5K is below the significance level of 0.05, the results suggest similar codebook usage across tasks, indicating code reuse for solving different tasks but with similar images.

## 6 ASAC Integration to Language Models

We tested the ASAC integration with a small NLP model using simple classification tasks. Training an NLP model from scratch is impractical due to the vast datasets and significant computational resources required. Thus, we use pre-trained DistilBERT model (Sanh et al., 2019) for NLP experiments. However, integrating the VQVAE module into all layers of such a pre-trained model presents challenges. Ideally, one would train only the new VQVAE parameters while keeping the pre-trained weights unchanged. However, adding untrained parameters to a pre-trained network disrupts the synergy between components, as the pre-trained weights are already optimized to work together, whereas the new VQVAE parameters have not undergone this collective training process. This misalignment causes the model to struggle to adapt, leading to poor performance.

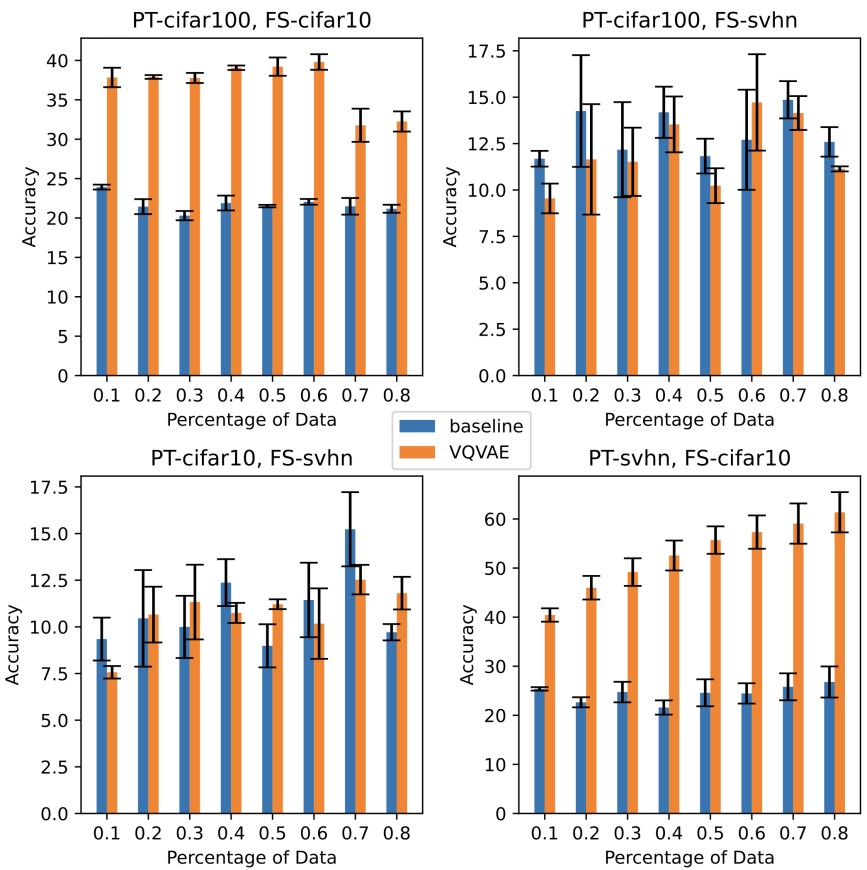

Figure 10: Results of few-shot experiments. PT and FS stands for datasets used for pre-training and few-shot fine-tuning.

To prevent disrupting the pre-trained weights, we incorporate the ASAC module solely into the last layer of the model, with its parameters initialized randomly. The baseline model retains pre-trained weights for all layers except the last one, which is initialized randomly. Similarly, the tested model maintains pre-trained weights for all layers but includes the ASAC module in the last layer. We then fine-tune both the baseline DistilBERT model and the ASAC-integrated DistilBERT. We test the model on GLUE[11] benchmark datasets.

### 6.0.1   Results

We show the results with the GLUE dataset in Table 2. We fine-tune the DistilBERT model for five different seeds and present the aggregate result. We observe that for all the experiments, the ASAC model outperforms the baseline. We also report the *p-value* to indicate the statistical significance of the differences between the VQVAE performance and the baseline. A lower *p-value* (typically less than 0.05) suggests that the differences are statistically significant and unlikely to be due to random variation, thereby indicating that the VQVAE integration has a meaningful impact on the model's performance. However, in some instances, the *p-value* exceeds 0.05, indicating that the results may not be statistically significant.

## 7   Discussion

In this paper, we delve into the integration of cognitive science principles into artificial intelligence (AI), specifically through the lens of Attention Schema Theory. We present Attention Schema-based Attention Controller, for more efficient control of attention particularly in transformers. ASAC is built on the premise

---

[11]https://huggingface.co/datasets/nyu-mll/glue

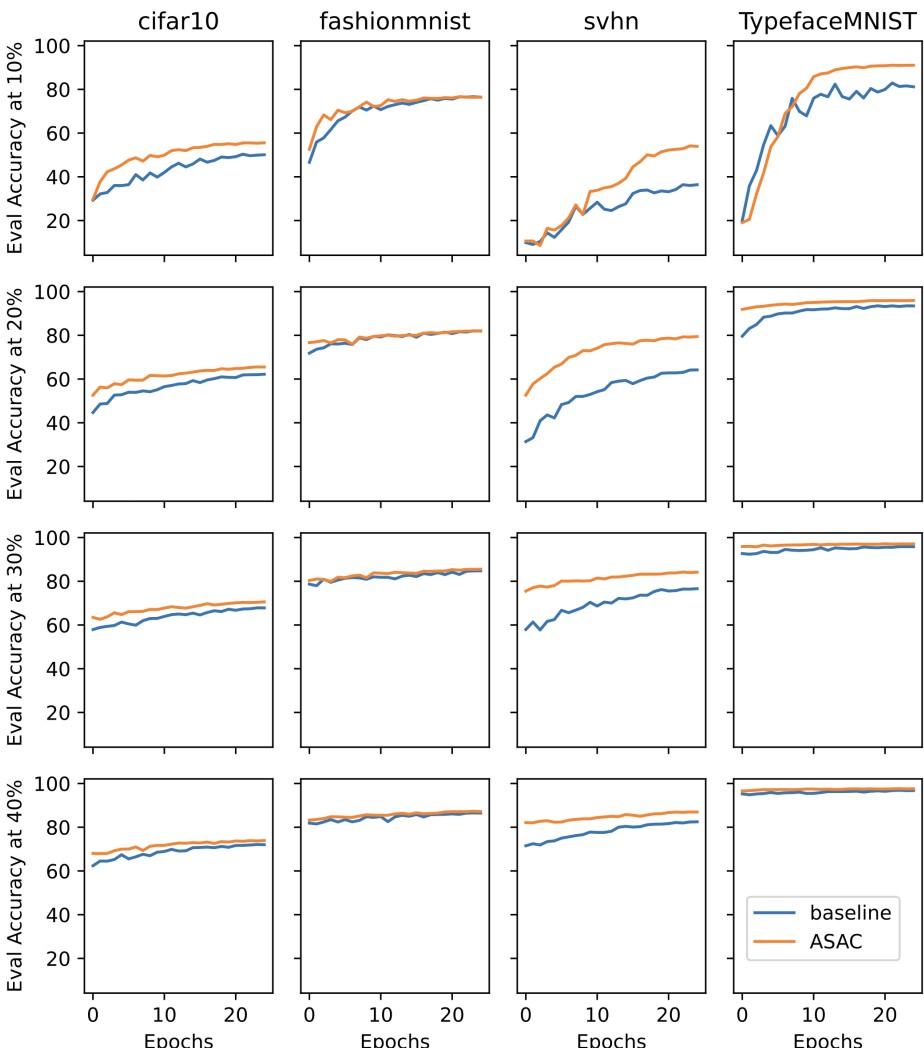

Figure 11: Results of learning efficiency experiments. Each row represents results for training the model with different percentages of data.

Table 2: Results with GLUE dataset

| GLUE Dataset | Metric | Baseline | ASAC | p-value |
|---|---|---|---|---|
| STSB | Pearson | $0.8644 \pm 0.0029$ | $\mathbf{0.8651 \pm 0.0012}$ | 0.5795 |
| | SpearmanR | $0.8622 \pm 0.0027$ | $\mathbf{0.8625 \pm 0.0010}$ | 0.8327 |
| SST2 | Accuracy | $0.8986 \pm 0.0102$ | $\mathbf{0.9052 \pm 0.0025}$ | **0.0001** |
| QNLI | Accuracy | $0.8850 \pm 0.0019$ | $\mathbf{0.8895 \pm 0.0027}$ | **0.0105** |
| MRPC | Accuracy | $0.8884 \pm 0.0011$ | $\mathbf{0.8915 \pm 0.0026}$ | **0.0311** |
| | F1 | $0.8367 \pm 0.0021$ | $\mathbf{0.8431 \pm 0.0073}$ | **0.022** |
| QQP | Accuracy | $0.9028 \pm 0.0008$ | $\mathbf{0.9033 \pm 0.001}$ | 0.057 |
| | F1 | $0.87 \pm 0.0013$ | $\mathbf{0.8705 \pm 0.0016}$ | **0.05** |
| RTE | Accuracy | $0.5870 \pm 0.0126$ | $\mathbf{0.6238 \pm 0.0131}$ | **0.0002** |

that human attention is an active process involving internal models that manage cognitive resources. According to the AST, individuals create simplified representations of their attention systems, enabling them to prioritize and allocate cognitive resources effectively. This concept has been translated into ASAC, which

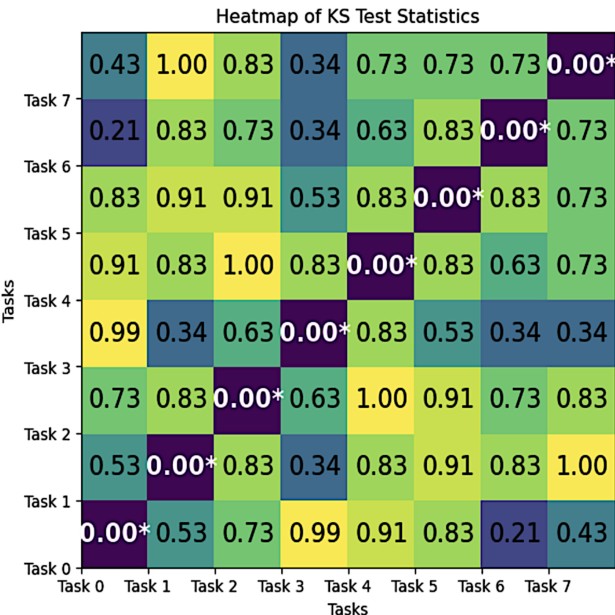

Figure 12: KS test *p-value* for ODIR-5K.

employs a Vector Quantized Variational AutoEncoder as both an attention abstractor and controller within transformer architectures.

Our experimental results across various datasets in vision and NLP domains highlight ASAC's effectiveness in improving classification accuracy and accelerating learning. For instance, when embedded within vision transformers, ASAC consistently outperformed baseline models across multiple datasets. These findings demonstrate that ASAC not only boosts classification accuracy but also expedites the learning process, showcasing its potential to enhance overall AI model performance.

One standout feature of ASAC is its robustness and generalization capabilities, particularly when exposed to noisy and out-of-distribution datasets. ASAC's ability to maintain performance under such challenging conditions underscores its adaptability and resilience—traits essential for real-world applications where data can often be unpredictable and varied.

Moreover, ASAC has shown promising results in multi-task settings, effectively managing attention across different tasks. This capability aligns with the human brain's cognitive flexibility, allowing seamless transitions between tasks and efficient resource allocation. Experimental results indicate that ASAC can dynamically adjust its attention allocation based on each task's requirements, a significant advancement over traditional attention mechanisms that often operate with fixed patterns.

Overall, with some preliminary experiments, we also demonstrate that the ASAC architecture is versatile and adaptable in many scenarios with some mixed results here and there. ASAC shows enhanced resilience to adversarial attacks. ASAC also excels in scenarios involving efficient learning and knowledge transfer. The model demonstrates improved performance in few-shot learning tasks, effectively learning from limited examples.

The integration of ASAC into language models, specifically the DistilBERT model, further demonstrates its potential for enhancing NLP tasks. Despite the challenges associated with incorporating the VQVAE module into pre-trained models, the results indicate that ASAC can improve performance on benchmark datasets like GLUE. This integration highlights ASAC's potential to enhance not only vision models but also language models, thereby broadening its applicability across different AI domains.

In future, we will address the current limitations of this study by effectively incorporating the attention controller into larger and pre-trained models like large language models (LLMs). Additionally, we aim to

explore and develop better architectures that more closely mimic human cognition. These efforts will not only enhance the performance and applicability of ASAC but also push the boundaries of how cognitive science principles can be integrated into advanced AI systems, ensuring they perform robustly and efficiently across various domains and tasks.

In conclusion, the development of ASAC represents a significant step forward in integrating cognitive science principles into AI. The implications of ASAC extend beyond mere performance improvements; they also shed light on the potential for creating AI systems that more closely mimic human cognitive processes. As AI continues to evolve, approaches like ASAC will be instrumental in developing models that can adapt, learn efficiently, and perform robustly in diverse and unpredictable environments.

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
