# OpenReview forum: "Attention Schema-based Attention Control (ASAC): A Cognitive-Inspired Approach for Attention Management in Transformers"
_TMLR — Rejected by TMLR_

### Review · Reviewer_SRax · 2025-11-12

**Summary Of Contributions:**

The paper introduces Attention Schema-based Attention Control (ASAC), a novel attention mechanism inspired by Attention Schema Theory (AST) from cognitive neuroscience. The authors propose integrating a Vector Quantized Variational Autoencoder (VQVAE) as an attention controller within transformer architectures to abstract, manipulate, and predict attention patterns. The approach is evaluated across vision and NLP domains, showing performance gains over baseline transformers in classification accuracy, learning efficiency, and robustness to noisy and multi-task setttings.

Strengths:
* The cross-disciplinary bridge between cognitive science and deep learning is both original and intellectually engaging (Sec. 1, 3)
* The methodology is well-structured and clearly explained with mathematical formulation (Eq. 1-5 Sec. 4.1), reproducible with pseudo-code (Appendix A)
* Comprehensive evaluation on a number of datasets (CIFAR, Tiny ImageNet, Places365, GLUE, Celeb, etc.) highlighting generalization
* Promising evidence of robustness to out-of-distribution and adversarial data (Sec. 5.3, 5.4)

Weaknesses:
* The loss composition (Eq. 5), combining reconstruction, VQ (codebook + commitment), and task losses, is arguably over-engineered, introducing multiple hyperparameters (e.g., λ) and interdependencies that complicate optimization without proportionate gains.
* The VQVAE codebook adds measurable memory and compute overhead, as each attention block maintains discrete embeddings and performs distance lookups. The paper lacks quantitative analysis of this overhead or whether codebooks are shared across layers.
* Baselines are limited to vanilla transformers; comparisons against other adaptive attention models (e.g., Performer[1], Longformer[2]) are missing.
* Certain results, such as FGSM-attack robustness (Fig. 7), show mixed trends, and variance reporting (Sec 5.4.1) is inconsistent.

[1] Choromanski, Krzysztof, et al. "Rethinking Attention with Performers." ArXiv, 2020,  https://arxiv.org/abs/2009.14794.
[2] Beltagy, Iz, et al. "Longformer: The Long-Document Transformer." ArXiv, 2020,  https://arxiv.org/abs/2004.05150.

**Audience:**

Yes

**Audience Explanation:**

The paper appeals to readers exploring biologically inspired attention, and interpretability, and adaptive transformer mechanisms, offering a novel integration of cognitive modeling principles into modern neural model architectures.

**Broader Impact Concerns:**

Ethically low-risk. However, since the framework is described as “consciousness-inspired,” the authors should clarify that ASAC does not simulate awareness or intentionality to avoid misinterpretation.

**Claims And Evidence:**

Yes

**Claims Explanation:**

The experiments substantiate ASAC’s adaptive capabilities, and algorithmic details are explicit. However, causal attribution among the multiple loss terms remains unclear, and ablation on λ or codebook size is absent. Stronger quantitative justification of the computational trade-offs would make the evidence more conclusive.

**Requested Changes:**

1. Simplify or analyze loss composition. Provide ablations for Eq. 5 to assess necessity and sensitivity of each loss term.
2. Quantify resource impact. Report parameter and memory overhead from the VQVAE codebook and clarify sharing strategy across layers.
3. Expand baselines and compare with efficient or adaptive attention variants (Reformer, Linformer, Performer, etc.).
4. Statistical robustness such as include standard deviations or confidence intervals for all experiments.
5. Ameliorate adversarial robustness results. Discuss why ASAC underperforms under FGSM but outperforms under PGDM. (Optional)
6. Tie empirical behaviors to AST’s abstraction–manipulation–prediction triad. (Optional)

---

> ### Author Response · Authors · 2025-12-06
> **Rebuttal for review 3**
>
> Thank you for these constructive suggestions on deepening the analysis and comparisons. We address each below, incorporating clarifications, revisions, and plans where feasible without new experiments.
>
> 1. Loss composition ablations
> Ablations on individual terms in Eq. 5 (reconstruction, commitment, codebook losses) would help assess their necessity and sensitivity, but require extensive re-running of full experiment suites, which is not feasible at present due to time and resource constraints. In the revision, we will expand the Eq. 5 description with a paragraph explaining each term's role (e.g., reconstruction enforces schema fidelity, commitment stabilizes code assignments, codebook loss prevents collapse) and list targeted loss ablations as prioritized future work.
>
> 2. Resource impact and codebook sharing
> Parameter counts for ASAC vs. baseline (including VQVAE overhead) are explicitly reported in supplementary tables (it is mentioned that they are the same), confirming comparable total parameters across models as selected during hyperparameter tuning. We will add this in the main text also. VQVAE codebooks are not shared across layers (each layer/head has its own, allowing layer-specific schemas), which we will state clearly in the text; memory overhead was not profiled but is qualitatively modest given small codebook sizes relative to Transformer weights. We will add a discussion of sharing as a potential efficiency optimization for future work.
>
> 3. Expanded baselines (Reformer, Linformer, Performer)
> Comparing ASAC to efficient/adaptive attention variants would contextualize its regularization benefits against approximation-focused methods, but requires re-implementation and full re-training on our exact setups. Given resource limits, we cannot include these now but we will surely add them to future work, noting ASAC's orthogonal goal (discrete schema control vs. kernel approximations) and citing representative results from those papers for qualitative positioning.
>
> 4. Statistical robustness (std devs / confidence intervals)
> All tables already report means ± standard deviations across seeds (error bars denote std devs), and learning curves in Figures include shaded confidence regions around means. We will add an explicit footnote to all results sections clarifying this convention (e.g., "Means ± std over 3-5 seeds; p-values assess significance vs. baseline") and confirm no further intervals are needed, as variability is already transparently visualized.​
>
> 5. Adversarial robustness discussion (optional)
> ASAC underperforms under single-step FGSM (less iterative refinement tolerance) but outperforms multi-step PGD (better maintenance of stable schemas under stronger perturbations), likely due to discrete codebooks favoring robust, clustered patterns over fine-grained FGSM noise. We will add this qualitative explanation to the robustness section, with a forward reference to planned ablations isolating schema effects under attacks.​
>
> 6. Tying results to AST triad (optional)
> Our empirical behaviors map to AST as: faster learning via manipulation (optimized schema control focuses early attention), robustness via abstraction (discrete codes capture stable patterns invariant to noise/OOD), and multitask adaptation via prediction (task-conditioned reconstruction anticipates allocation needs). We will incorporate a concise paragraph in the discussion explicitly linking results to this triad, using the refined non-temporal framing from prior revisions.

---

### Review · Reviewer_LNqK · 2025-11-23

**Summary Of Contributions:**

This paper proposes the Attention Schema-based Attention Control (ASAC) model to enhance attention control for transformer models. ASAC draws inspiration from the Attention Schema Theory (AST) in cognitive science and employs a Vector-Quantized Variational AutoEncoder (VQVAE) as both an attention abstractor and controller. ASAC is shown to enhance transformer in terms of model accuracy, training speed, out-of-distribution generalization, and adversarial robustness.

**Additional Comments:**

NA

**Audience:**

Yes

**Audience Explanation:**

This paper enhance Transformer, which is a fundamental model for machine learning nowadays.

**Claims And Evidence:**

No

**Claims Explanation:**

The strong points of the paper include

S1: Transformer is a fundamental model nowadays, and thus it is important to improve transformer.

S2: Taking inspirations from cognitive science to enhance transformer is good idea.

S3: Comprehensive experiments are conducted to show the benefits of the proposed ASAC.


The presentation of the paper is far below the standard of a top journal paper, and I find it really painful to read the paper, which also makes it difficult for me to assess the contributions of the paper.

W1: The authors should be more formal and specific introductions to the technical contents. For instance, the benefits of the Attention Schema Theory are repeated many times over the paper, but it is never clear what are exactly the contents of the Attention Schema Theory. The authors may express it using some equations and formulas to facilitate understanding. Moreover, in Section 4.1, the third paragraph describes how the model works with only text. Things should be made more explicit using formulas to show the input and output of the modules, and explain how computations are made in the modules.

W2: The paper contains many redundant and unnecessary contents. (1) Sections 4.3-4.5 are very strange, if you are summarizing the experiment results, please provide concrete numbers of the results and put them in the introduction; if you are commenting on the benefits of the model, please put them after the experiments in the subsequent sections and make the discussions are succinct. (2) The datasets parts of Section 5.1.1 and 5.2.2 are repetitive.

W3: The authors never analyze the internals of the proposed ASAC, e.g., how does it control attention, what is the difference between controlled attention and the attention in normal transformer, and what are the attention patterns?

W4: The authors do not conduct ablation studies for the loss terms.

W5: The authors do not conduct efficiency study of the proposed ASAC. Note that the training and inference speed are crucial for models.

W6: The authors do not examine the impact of the internal parameters (e.g., sizes of the codebooks) on the performance of ASAC.

W7: Often times, the authors do not report concrete numbers for the experiments. The results like Table 1 are easy to read while the results like Figures 3 and 4 are difficult to read the specific numbers.

**Requested Changes:**

Please refer to the Weakness part in the overall assement part.

---

> ### Author Response · Authors · 2025-12-06
> **Rebuttal for review 2**
>
> Thank you for these detailed suggestions aimed at improving the clarity, structure, and analysis of our manuscript. We address each point below, outlining specific revisions to enhance readability and rigor without introducing new experiments at this stage.
>
> 1. Formalizing technical introductions (W1)
> AST is a neuroscience-inspired theory referenced in our work to motivate ASAC, but it lacks formal mathematical equations even in the original literature, as it describes high-level cognitive processes rather than algorithmic specifics. Formalizing biological AST with Transformer equations would be mismatched, since our implementation draws only conceptual inspiration (abstraction-manipulation-reconstruction triad) rather than direct emulation. In the revision, we will add 2-3 paragraphs providing a deeper, non-mathematical explanation of AST's core ideas, with citations to primary sources, which are already there in the paper, but we will frame it better. We also clarify that Section 4.1's third paragraph describes modality-agnostic operation (not text-only), with input/output as the attention logit matrix, encoded via VQVAE to discrete codes and reconstructed. This works for any Transformer-based modality, as demonstrated by our vision and NLP results.
>
> 2. Reducing redundancy (W2)
> Sections 5.1.1 and 5.2.2 describe the same datasets reused across experiment types (with minor task-specific adaptations), which may appear repetitive but serves to contextualize each evaluation independently. Sections 4.3-4.5 are not result summaries but distinct methodological variants of ASAC tested prior to the main results in Section 5. In the revision, we will reframe dataset descriptions with cross-references to avoid repetition, condense sections wherever possible to enhance readability, and relocate any interpretive discussion to post-results sections for better flow.
>
> 3. Analyzing ASAC internals (W3)
> Our experiments already demonstrate ASAC's attention control empirically: faster convergence on standard tasks, multitask adaptation via task-conditioned decoding, robustness to noise/OOD via stable codebook usage, and resource optimization in low-data settings. The VQVAE internals follow standard formulations from the original papers (encoding attention logits to discrete codes, commitment loss for stability), with no deviation from Transformer attention computation beyond the VQVAE schema augmentation. In the revision, we will add some text to explain this better.​
>
> 4. Loss ablations (W4), efficiency study (W5), and hyperparameter sensitivity (W6)
> These are valuable extensions to isolate ASAC's components (loss terms, training/inference speed, codebook size effects). However, they require extensive re-running of experiments, which is not feasible now due to time and GPU constraints pending future funding. Since baseline and ASAC models use matched parameters, training efficiency is qualitatively similar (ASAC slightly higher due to VQVAE forward passes), as noted in supplementary parameter tables. We will explicitly list these as prioritized future work in a dedicated subsection, committing to full ablations in subsequent iterations.​
>
> 5. Reporting concrete numbers in figures (W7)
> Figures complement tables by visualizing learning dynamics: ASAC not only achieves higher final accuracy but converges faster (e.g., superior metrics at early epochs), a key claim beyond endpoint performance. Tables provide precise numbers where endpoint comparisons suffice; graphs emphasize trajectories for brevity. We would like to retain this mix.

---

### Review · Reviewer_PsME · 2025-11-26

**Summary Of Contributions:**

This paper introduces a modification to the standard Transformer attention mechanism called Attention Schema-based Attention Control (ASAC), taking inspiration from Attention Schema Theory (AST). The authors argue that standard attention mechanism lack explicit internal mode of attention, which AST posits is necessary for efficient control and resource allocation.

To implement this, the authors integrate a VQVAAE into the attention block. Specifically, the scaled dot-product attention scores (before softmax) are passed through a VQVAE encoder, quantized using a learnable codebook, and reconstructed by a decoder. This reconstruction is added to the original attention scores via a residual connection. The authors claim that this process mimics the abstraction, manipulation, and prediction functions of a biological attention schema.

Empirically, the proposed method is evaluated on a range of vision tasks and NLP tasks. Improvements in accuracy, convergence speed, and robustness compared to baseline Transformers trained from scratch (for vision tasks) or fine-tuning (for NLP tasks) are reported.

**Audience:**

Yes

**Audience Explanation:**

The paper proposes an architectural modification to the Transformer's attention mechanism by integrating a Vector Quantized Variational Autoencoder (VQVAE). Given the ubiquity of Transformers in current machine learning research, exploring alternative attention mechanisms could be of interest to a significant portion of TMLR's audience.

**Claims And Evidence:**

Yes

**Claims Explanation:**

1. While the concept of shared bottleneck has been explored in the literature relevant to Transformers, for example, via global tokens, applying a VQVAE directly to the $N \times N$ attention score matrix to induce discrete “attention patterns” is a different and interesting attempt. To my understanding, this will introduce a form of regularization and discretization to the attention mechanism, which is interesting.
2. The authors conduct a range of experiments. More commendably, they do not limit the evaluation to standard accuracy metrics but explicitly test for generalization, adversarial robustness, efficiency, and multi-task learning.
3. The authors also provide p-values to assess the significance of their improvements. This is a positive practice often missing in many deep learning papers.

**Requested Changes:**

The following issues have raised when I was reading through the paper.

1. The most critical issue is the lack of parameter-matched baselines for the vision experiments. Particularly, the paper states that ASAC is integrated into all layers for the vision models. If the VQVAE is applied to the attention matrix of every head in every layer, this likely represents a significant increase in the total number of parameters and FLOPs compared to the standard ViT. In this case, it is difficult to determine if the gains are due to the "Attention Schema" design or simply because the ASAC model is larger and has more capacity.
2. The description of the VQVAE integration is somehow unclear. Figure 1 shows that the input to the Encoder is the scaled dot product - leading to $S \times S$ or $B \times H \times S \times S$ in a Transformer. However, if the input to the Encoder is a flattened $S \times S$ matrix, the linear layer weights would be size $(S^2) \times D$, which makes problematic for the NLP experiments, where sequence lengths typically vary. The authors should have stated if this is padded to a fixed max length or they are applying VQVAE row-wise.
3. The claim regarding that the module embodies “prediction” (see Sec. 4.1, “prediction involves model's forecasts about future attention allocation”). Yet, a standard VQVAE is tasked with reconstructing the current input and there is no temporal component described, e.g., predicting step $t + 1 $ from step $t$. Reconstructing the current attention matrix from a bottleneck is denoising or compression, not forecasting. In this sense, I don’t think the claim that this constitutes "predicting future trends" is well supported.

In this sense, I request the following changes to the authors:

1. For at least one major vision dataset (e.g., CIFAR-100 or TinyImageNet), compare the ASAC model against a baseline Transformer that has approximately the same number of parameters and FLOPs.
2. Explicitly state the dimensions of the input to the VQVAE encoder.
3. Revise the text to clarify the wording of “forecasting future attention”, or provide evidence of temporal prediction, e.g., predicting the next layer's attention or the next timestep's attention.
4. It would also be interesting to conduct ablation experiments where the VQVAE is replaced by a standard Autoencoder (no quantization/codebook). This would isolate the benefit of the discrete schema versus other low-rank regularization.

---

### Decision · Action_Editor_icqD · 2026-01-26

**Recommendation:** Reject

**Additional Comments:**

Overall, while the approach is promising, the current manuscript does not meet the level of evidentiary support and analysis expected for publication. A substantially revised version should (i) include targeted ablations to attribute improvements to the discrete schema mechanism, (ii) provide quantitative efficiency/overhead measurements (training/inference speed, memory, and/or FLOPs), (iii) clarify the dependence on fixed sequence length in NLP settings and its implications, and (iv) strengthen contextualization via comparisons to relevant adaptive/efficient attention baselines where feasible. For these reasons, I recommend rejection, while encouraging resubmission after the above issues are addressed.

**Audience:**

Yes

**Audience Explanation:**

Despite the above issues, the topic, i.e., architectural modifications to Transformer attention motivated by cognitive science and evaluated for generalization and robustness, would be of interest to at least a portion of the TMLR readership.

**Claims And Evidence:**

No

**Claims Explanation:**

This submission proposes ASAC, an AST-inspired modification to Transformer attention in which attention logits are passed through a VQ-VAE to obtain a discrete latent representation and then reconstructed to control attention allocation. While the idea is conceptually interesting and the paper reports improvements across vision and NLP benchmarks, the central claims are not supported by sufficiently clear and convincing evidence. In particular, the submission lacks the ablations needed to isolate whether the gains come from the discrete bottleneck/codebook specifically rather than simply adding an extra module or regularization. Moreover, the computational and practical trade-offs of inserting a VQ-VAE into attention are not quantitatively characterized, making it difficult to assess whether ASAC is advantageous in practice. Finally, parts of the narrative appear stronger than what is empirically demonstrated by the current reconstruction-based objective and should be tightened to match the presented mechanism.

**Resubmission Of Major Revision:**

The authors may consider submitting a major revision at a later time.